# Vasculoprotective Potential of Baicalein in Angiotensin II-Infused Abdominal Aortic Aneurysms through Inhibiting Inflammation and Oxidative Stress

**DOI:** 10.3390/ijms242116004

**Published:** 2023-11-06

**Authors:** Erna Sulistyowati, Shang-En Huang, Tsung-Lin Cheng, Yu-Ying Chao, Chia-Yang Li, Ching-Wen Chang, Meng-Xuan Lin, Ming-Chung Lin, Jwu-Lai Yeh

**Affiliations:** 1Faculty of Medicine, University of Islam Malang, Malang City 65145, Indonesia; dr_erna@unisma.ac.id; 2Department of Pharmacology, College of Medicine, Kaohsiung Medical University, Kaohsiung 807, Taiwan; eva_1433@yahoo.com.tw (S.-E.H.); bay1247@gmail.com (C.-W.C.); g0925317135@gmail.com (M.-X.L.); 3Graduate Institute of Medicine, College of Medicine, Kaohsiung Medical University, Kaohsiung 807, Taiwan; chiayangli@kmu.edu.tw; 4Department of Physiology, College of Medicine, Kaohsiung Medical University, Kaohsiung 807, Taiwan; junglecc@gmail.com; 5Regenerative Medicine and Cell Therapy Research Center, Kaohsiung Medical University, Kaohsiung 807, Taiwan; 6College of Professional Studies, National Pingtung University of Science and Technology, Pingtung 912, Taiwan; 7Department of Public Health, College of Health Sciences, Kaohsiung Medical University, Kaohsiung 807, Taiwan; yuyich@kmu.edu.tw; 8Department of Anesthesiology, Chi Mei Medical Center, Tainan 710, Taiwan; 9Department of Medical Laboratory Science and Biotechnology, Chung Hwa University of Medical Technology, Tainan 717, Taiwan; 10Department of Medical Research, Kaohsiung Medical University Hospital, Kaohsiung 807, Taiwan; 11Department of Marine Biotechnology and Resources, National Sun Yat-sen University, Kaohsiung 804, Taiwan

**Keywords:** baicalein, vasculoprotective, abdominal aortic aneurysm, inflammation, oxidative stress

## Abstract

Aortic wall inflammation, abnormal oxidative stress and progressive degradation of extracellular matrix proteins are the main characteristics of abdominal aortic aneurysms (AAAs). The nucleotide-binding oligomerization domain-like receptor family pyrin domain containing 3 (NLRP3) inflammasome dysregulation plays a crucial role in aortic damage and disease progression. The first aim of this study was to examine the effect of baicalein (5,6,7-trihydroxy-2-phenyl-4H-1-benzopyran-4-one) on AAA formation in apolipoprotein E-deficient (ApoE^−/−^) mice. The second aim was to define whether baicalein attenuates aberrant vascular smooth muscle cell (VSMC) proliferation and inflammation in VSMC culture. For male ApoE^−/−^ mice, a clinically relevant AAA model was randomly divided into four groups: saline infusion, baicalein intraperitoneal injection, Angiotensin II (Ang II) infusion and Ang II + baicalein. Twenty-seven days of treatment with baicalein markedly decreased Ang II-infused AAA incidence and aortic diameter, reduced collagen-fiber formation, preserved elastic structure and density and prevented smooth muscle cell contractile protein degradation. Baicalein inhibited rat VSMC proliferation and migration following the stimulation of VSMC cultures with Ang II while blocking the Ang II-inducible cell cycle progression from G_0_/G_1_ to the S phase in the synchronized cells. Cal-520 AM staining showed that baicalein decreased cellular calcium in Ang II-induced VSMCs; furthermore, a Western blot assay indicated that baicalein inhibited the expression of PCNA and significantly lowered levels of phospho-Akt and phospho-ERK, along with an increase in baicalein concentration in Ang II-induced VSMCs. Immunofluorescence staining showed that baicalein pretreatment reduced NF-κB nuclear translocation in Ang II-induced VSMCs and furthered the protein expressions of NLRP3 while ASC and caspase-1 were suppressed in a dose-dependent manner. Baicalein pretreatment upregulated Nrf2/HO-1 signaling in Ang II-induced VSMCs. Thus, 2′,7′-dichlorodihydrofluorescein diacetate (DCFH-DA) staining showed that its reactive oxygen species (ROS) production decreased, along with the baicalein pretreatment. Our overall results indicate that baicalein could have therapeutic potential in preventing aneurysm development.

## 1. Introduction

Abdominal aortic aneurysm, a progressive pathological vascular dilatation, is considered a chronic inflammatory degenerative aortic disease which primarily affects the elderly [1]. Age [2], male gender [3], smoking [3], hypertension [3], possible genetic susceptibility [4] and dyslipidemia [5] are common risk factors for this disease. Because aneurysms of the abdominal aorta are usually asymptomatic, the current clinical challenge is to diagnose them in their early stages and elucidate the biological mechanisms responsible for their progressive dilatation and eventual rupture, creating new diagnostics and therapeutic approaches. 

Although the underlying mechanism of the development and progression of AAAs is not completely understood, inflammation plays an important role [6]. Environmental and genetic risk factors lead to the development of aortic atherosclerosis, while resultant positive remodeling, intimal thrombosis and release of pro-inflammatory cytokines stimulate secondary matrix degradation and adventitial inflammation that promotes AAA development [7]. The main mechanisms involved are the loss of vascular smooth muscle cells through apoptosis [7,8], accelerated degradation of the extracellular matrix caused by increased expression of metalloproteinases (but a decreased number of tissue inhibitors) [9], increased renin–angiotensin system levels and activity [10,11], enhanced oxidative tissue damage [12,13,14], dysregulation of pro- and anti-inflammatory mediators generated from immune cells, enhanced angiogenesis of the mural [15,16], altered hemodynamics and homeostasis [8,17] and increased genetic susceptibility [5,18]. Certain clinical studies have shown that inflammatory cell infiltration is a common feature seen in the arterial wall of AAA patients [19].

Recent research suggests that inflammation in AAAs is predominantly mediated by the NLRP3 inflammasome [19], which is a cytosolic multiprotein complex that activates caspase-1 and controls the production of pro-inflammatory cytokines such as interleukin (IL)-1 and IL-18 as well as the stimulation of lytic cell death, known as pyroptosis, and thereby causes inflammation. Previous research has shown that the NLRP3 inflammasome and the downstream cytokine IL-1 play an important role in the development of AAAs [20]. This shows that NLRP3-driven IL-1 might be a therapeutic target for AAAs. The NLRP3 inflammasome is a well-studied inflammasome with three primary components: the NLRP3 receptor protein, the ASC adapter protein and the caspase-1 effector protein [21]. The NLRP3 protein has one leucine-rich repeat for ligand recognition, one oligomerization domain for nucleotide binding and one pyrin domain for ASC binding [22], while one pyrin domain and one caspase recruitment domain are found in the ASC protein, which attaches to the NLRP3 protein via its pyrin domain and recruits and activates caspase-1 via its caspase recruitment domain once the NLRP3 inflammasome is active. Two signaling models for NLRP3 inflammasome activation have now been proposed. Signal 1 is given by microbial components or endogenous cytokines, resulting in the activation of the transcription factor nuclear factor-κB (NF-κB) and subsequent upregulation of NLRP3 and pro-interleukin-1 (pro-IL-1) [23]; herein, caspase-8 and FAS-mediated death domain protein (FADD) as well as NOD1/2 are involved in priming by modulating the NF-κB pathway where NLRP3 undergoes post-translational changes that allow it to be activated. A number of stimuli, including extracellular ATP, pore-forming toxins, RNA viruses and particulate matter, produce the activation signal, or signal 2 where multiple molecular or cellular processes have been found to activate the NLRP3 inflammasome, including ionic flux, mitochondrial malfunction and ROS production and lysosomal degradation [24]. 

Many clinical and experimental studies have demonstrated elevated levels of NLRP3 inflammasome effector mediators such as IL-1β and IL-18 in AAAs [25,26] with the major sources of the increased expression of these effectors being aortic macrophages and vascular smooth muscle cells. In addition, a large body of evidence from both clinical and experimental studies has shown that increased expression of NLRP3 inflammasome components; for instance, ASC, caspase-1 and NLRP3, is a relevant indicator of cardiovascular disease severity [21]. Taken together, accumulated evidence from recently published research supports the significant finding that the NLRP3 inflammasome is a key driver of AAAs.

Baicalein, a flavonoid originally isolated from the roots of *Scutellaria baicalensis* Georgi, has been reported to possess antibacterial, antiviral and specifically anti-inflammatory activities in vitro and in vivo in mouse models [27]. *Scutellaria baicalensis* Georgi is a prominent herbal remedy in traditional Chinese medicine that is utilized in China and other Asian nations. Recent studies have shown that baicalein has various pharmacological properties, including antioxidant benefits and cardiovascular preventive effects through suppression of the NF-κB pathway [27,28]. The only acceptable treatment for AAAs nowadays is surgery. There is typically a long gap between AAA diagnosis and corrective surgery, during which an efficient medication therapy might prevent or postpone the necessity for AAA repair. Experimental in vivo and in vitro studies suggest the potential benefits of baicalein in preventing or reducing the progression of AAAs [29,30]; moreover, this extensively utilized natural product has been demonstrated to work via a variety of processes, including gene expression control, metabolism and cell signaling pathways [30]. Regarding the causative role of the NLRP3 inflammasome in the pathogenesis of AAAs, in the present study, we aimed to determine whether baicalein prevents the development of experimental AAAs through an in vivo experiment. To explore baicalein’s effects the attenuation of AAAs during VSMC growth, we generated Ang II-induced VSMC cultures to define whether baicalein inhibits cell proliferation, cell motility, cell migration, cell cycle, Ca^2+^ release and PCNA, ERK and Akt activation. The protein expressions of NLRP3 and relative nuclear NF-κB intensity were defined in Ang II-induced VSMCs to demonstrate that baicalein inhibits inflammation. The protein expressions of Keap1, Nrf2 and HO-1 were evaluated to determine whether baicalein decreases ROS generation.

## 2. Results

### 2.1. Baicalein Treatment Effectively Limits Ang II-Infused Abdominal Aortic Aneurysm Formation

To investigate the therapeutic effects of baicalein in AAA formation, we generated an in vivo study using the Ang II-infused AAA model in mice. On day 28, no rupture-associated mortality was observed in any of the groups; however, the maximal diameter of the suprarenal aorta was significantly increased in the AAA group (1.42 ± 0.09 mm) compared with the sham group (0.58 ± 0.02 mm; *p* < 0.001; Figure 1a,b), and the diameter became significantly smaller in the AAA + BE group (0.82 ± 0.03 mm) than in the Ang II group (*p* < 0.001) while the baicalein group had a diameter of 0.59 ± 0.03 mm (*p* < 0.001), which is smaller than that of the Ang II group. Our findings show that focal enlargement of the suprarenal aorta was induced by Ang II infusion while baicalein significantly inhibited the Ang II-induced increase in the suprarenal aorta diameter. Due to the fact that baicalein prevented vascular fibrosis, we analyzed the excess deposition of collagen induced by AAAs. As shown in Figure 1c,d, compared with the AAA group (24.39 ± 1.03%), baicalein treatment significantly reduced the percentage of positive collagen area (13.94 ± 0.96%, *p* < 0.001) in the Ang II-infused AAA group. AAAs commonly cause VSMC loss and then lead to medial vascular remodeling [31], while our histological examination demonstrated that baicalein significantly inhibited Ang II-induced excessive collagen deposition of the suprarenal aorta with the collagen density being lower in baicalein-treated mice than Ang II-stimulated mice. As shown in Figure 1e,f, in the medial layer, α-SMA staining was scarcely observed in the Ang II group, leading to a substantial reduction in the overall α-SMA staining-positive area in the Ang II group (10.36 ± 0.83%, *p* < 0.001) as compared with the sham group (20.68 ± 1.46%). However, in the AAA + BE group, it was found that the decline in medial α-SMA staining-positive area was significantly attenuated by baicalein treatment (15.57 ± 0.70%, *p* < 0.01), suggesting that the baicalein treatment mitigated the quantitative reduction in VSMC content during aneurysm formation. In addition, baicalein was associated with better preservation of medial elastin; the mean grade of elastin degradation in the AAA + BE group (3.0 ± 0.57%, *p* < 0.001) was approximately half that in Ang II-infused AAA mice (11.75 ± 0.49%) (Figure 1g,h).

### 2.2. Baicalein Effectively Inhibits Ang II-Induced VSMC Proliferation, Motility and Migration

To test the biological function of baicalein in rat aortic VSMCs, we first analyzed the cytotoxicity effects of baicalein. After the treatment with various concentrations of baicalein (0, 1, 5, 10, 50, 100 μM) at three time points, the results of the MTT assay indicated that VSMC viability was not inhibited by baicalein (Figure 2a). The observation of cell density with a microscope and counting cell numbers via trypan blue staining and determination of cell proliferation via an XTT assay revealed that 10, 50 and 100 μM baicalein significantly attenuated the Ang II-induced increase in cell number (Figure 2b, *p* < 0.05). The percentages of cell proliferation in the control and Ang II, 1, 10, 50 and 100 μM baicalein groups were 100.0 ± 3.59, 117.5 ± 2.39, 111.7 ± 2.65, 103.56 ± 1.65, 101.83 ± 2.24 and 100.0 ± 2.18% respectively. After producing a scrape wound, untreated rat VSMCs migrated across the wound edge to fill in the area that was denuded. As shown in Figure 2c, the migration of VSMCs was promoted by treatment with Ang II (0.1 μM), while 1, 10 50 and 100 μM baicalein treatments significantly reduced cell migration in a concentration-dependent manner compared with the Ang II group. As shown in Figure 2e, the wound area coverage in the control and Ang II, 1, 10, 50 and 100 μM baicalein groups was 20.0 ± 1.38, 100.0 ± 2.55, 80 ± 2.27, 40.0 ± 2.04, 21.77 ± 1.2 and 21.6 ± 0.99%, respectively. The wound size of VSMCs was reduced by baicalein treatment as compared with Ang II-induced cells (*p* < 0.01). Furthermore, in order to confirm the inhibitory effect of baicalein on VSMC migration (Figure 2d), a Transwell Boyden chamber experiment was conducted. The number of VSMCs that migrated through the Transwell chamber was greatly increased by treatment with Ang II for 24 h (400.3 ± 13.12 cells, *p* < 0.001). Despite Ang II stimulation, the numbers of migrated cells were significantly reduced by 10, 50 and 100 μM baicalein treatments (Figure 2f), with the cell counts in the 1, 10, 50 and 100 μM baicalein treatments being 371.8 ± 16.81, 292.4 ± 27.41, 114.5 ± 5.69 and 98.23 ± 7.99 respectively, with all migration responses being significantly reduced in a dose-dependent manner (*p* < 0.01).

### 2.3. The Influence of Baicalein on Cell Cycle Progression of Ang II-Induced Cells

The influence of baicalein treatment on the cell cycle distribution of Ang II-stimulated VSMCs was analyzed via flow cytometry (Figure 3a). Serum deprivation of VSMCs for 24 h caused 81.5 ± 1.2% synchronization to the G_0_/G_1_ phase of the cell cycle (Figure 3b). The percentage of cells in the G_0_/G_1_ phase was reduced by Ang II stimulation in VSMCs (69.43 ± 2.87%, *p* < 0.01) compared with the control (81.48 ± 1.22%). Pretreatment with 10, 50 and 100 μM baicalein significantly increased the percentage of cells in the G_0_/G_1_ phase (*p* < 0.05). As shown in Figure 3c, the percentage of cells in the S phase was increased upon stimulation with Ang II (7.5 ± 0.5%, *p* < 0.001) compared with the control (7.53 ± 0.47%).

Pretreatment with 10, 50 and 100 μM baicalein significantly increased the percentage of cells in the S phase (*p* < 0.001). As shown in Figure 3d, the percentage of cells in the G_2_/M phase was increased by Ang II stimulation (16.9 ± 0.4%, *p* < 0.001) compared with the control (11.18 ± 0.88%). 

Pretreatment with 1, 10, 50 and 100 μM baicalein led to a non-significant increase in the percentage of cells in the G_2_/M phase. These data show that cell proliferation could be activated by Ang II. The reduction in the S-phase population and the increase in the fraction of cells in the G_0_/G_1_ phase among Ang II-treated cells occurred upon treatment with 10, 50 and 100 μM baicalein (Figure 3b,c). In contrast, baicalein treatment had a non-significant influence on the G_2_/M phase (Figure 3d). These data indicate that S-phase entry in Ang II-induced VSMCs might be arrested by baicalein in a dose–response relationship (*p* < 0.001).

### 2.4. Baicalein Effectively Reduces Ca^2+^ Release and the Activation of PCNA, ERK and Akt in Ang II-Induced VSMCs

To further confirm the regulatory effect of baicalein treatment on calcium deposition and the biomarkers of VSMC proliferation, VSMC cultures were confirmed via fluorescent staining for an intracellular calcium assay and Western blotting for protein expression analysis. Moreover, determination of the release of intracellular calcium via Cal-520 AM staining in primary VSMCs revealed that Ang II stimulation increased the release of intracellular Ca^2+^ (249.3 ± 12.5%), while this was partially reversed by treatment of baicalein (Figure 4b, *p* < 0.001). The percentages of fluorescent intensity in 1, 10, 50 and 100 μM baicalein were 200.1 ± 4.61, 137.0 ± 5.23, 107.2 ± 1.54 and 106.3 ± 2.55% respectively. These results verify the role of baicalein in the inhibition of Ang II-induced intracellular Ca^2+^ release. Previous studies demonstrated that PCNA, ERK and Akt have a considerable influence on the proliferation of Ang II-induced VSMCs [32,33]. As shown in Figure 4c,d, Ang II stimulation significantly upregulated the expression of PCNA (1.77 ± 0.05 relative units, *p* < 0.001). The levels of PCNA expression were 1.29 ± 0.12, 1.11 ± 0.04 and 0.74 ± 0.07 relative units (*p* < 0.001), respectively, after 10, 50 and 100 μM baicalein treatments. 

PCNA expression was reduced by the baicalein treatment. Similarly, Ang II stimulation significantly elevated p-ERK expression (4.3 ± 0.12 relative units, *p* < 0.001). As shown in Figure 4f, pretreatment with 10, 50 and 100 μM baicalein reduced p-ERK expression as follows: 3.73 ± 0.14, 1.28 ± 0.06 and 0.96 ± 0.04 relative units (*p* < 0.05), respectively. Likewise, Ang II stimulation significantly elevated the expression of p-Akt in VSMCs (1.61 ± 0.07 relative units, *p* < 0.001). The p-Akt expressions was downregulated after treatment with 10, 50 and 100 μM baicalein as follows: 1.12 ± 0.04, 1.0 ± 0.07 and 0.89 ± 0.06 relative units, *p* < 0.001, respectively (Figure 4h). These results demonstrate that baicalein treatment inhibited Ang II-induced VSMC proliferation and activation of ERK/Akt.

### 2.5. Baicalein Effectively Inhibits NF-κB and NLRP3 Inflammasome Pathway in Ang II-Induced VSMCs

To demonstrate the effectiveness of baicalein at inhibiting inflammation, VSMCs were exposed to Ang II, a well-characterized activator of NF-κB [34]. Immunofluorescence using an anti-p-NF-κB antibody was then employed to examine the effects of baicalein on the nuclear localization of p-NF-κB. Unstimulated VSMCs demonstrated cytoplasmic distribution of p-NF-κB, with the intensity of nuclear NF-κB being 100.0 ± 8.59%, while exposure of these cells to Ang II caused movement of p-NF-κB to the nuclear area, where the nuclear NF-κB intensity was 288.0 ± 16.39% (*p* < 0.001). This effect could be blocked by baicalein (50 μM), demonstrating its effectiveness at inhibiting NF-κB translocation (Figure 5a,b), where the nuclear NF-κB intensity was 140.0 ± 11.08% after 50 μM baicalein treatment (*p* < 0.001). Furthermore, the NLRP3 inflammasome is a multiprotein complex consisting of NLRP3, ASC and caspase-1 [24]. In order to understand how baicalein mediates NLRP3 component expression in Ang II-induced VSMCs, the protein levels of NLRP3 components were evaluated in VSMCs. As shown in Figure 5d, the protein level of NLRP3 was upregulated by Ang II stimulation (1.71 ± 0.11 relative units, *p* < 0.001). Pretreatment with 10 and 50 μM baicalein reduced NLRP3 expression as follows: 0.93 ± 0.06 and 0.77 ± 0.05 relative units (*p* < 0.001), respectively. As shown in Figure 5e, the level of ASC was elevated via Ang II stimulation (3.02 ± 0.11 relative units, *p* < 0.001). Pretreatment with 1, 10 and 50 μM baicalein reduced ASC expression as follows: 1.96 ± 0.13, 1.76 ± 0.09 and 0.73 ± 0.05 relative units (*p* < 0.001), respectively. As shown in Figure 5f, the level of caspase-1 was elevated by Ang II stimulation (1.79 ± 0.07 relative units, *p* < 0.001). Pretreatment with 1, 10 and 50 μM baicalein reduced caspase-1 expression as follows: 1.43 ± 0.08, 1.38 ± 0.07 and 0.67 ± 0.04 relative units (*p* < 0.05), respectively. Likewise, as shown in Figure 5g, the level of IL-1β was elevated by Ang II stimulation (1.31 ± 0.11 relative units, *p* < 0.05). Pretreatment with 1, 10 and 50 μM baicalein downregulated the protein levels of IL-1β as follows: 0.96 ± 0.07, 0.72 ± 0.05 and 0.51 ± 0.04 relative units (*p* < 0.05), respectively. These results imply that baicalein has an association with the suppression of NLRP3 components in Ang II-induced VSMCs in a dose-dependent manner.

### 2.6. Baicalein Effectively Induces of Nrf2/HO-1 Pathway against ROS Production in Ang II-Induced VSMCs

The upregulation of antioxidant activity caused by the Nrf2/HO-1 pathway plays a critical role in AAAs [35]. Upon activation, Nrf2 is an essential transcription factor and it regulates the expressions of antioxidant factors, thereby, exerting the protective effects. Furthermore, Nrf2 transcription is negatively regulated by Keap1, a key repressor of Nrf2. The results indicated that there was an upregulation of Keap1 expression in Ang II stimulation (1.94 ± 0.14 relative units, *p* < 0.001). As shown in Figure 6b, pretreatment with 10 and 50 μM baicalein reduced Keap1 expression as follows: 1.11 ± 0.06; and 0.93 ± 0.05 relative units (*p* < 0.001), respectively. As shown in Figure 6c, the level of Nrf2 was decreased by Ang II stimulation (0.3 ± 0.03 relative units, *p* < 0.001). Pretreatment with 1, 10 and 50 μM baicalein increased the levels of Nrf2 as follows: 0.72 ± 0.04; 0.72 ± 0.04; and 0.69 ± 0.04 relative units (*p* < 0.001) respectively. As shown in Figure 6d, the expression HO-1 in Ang II-induced VSMCs was 0.73 ± 0.03 relative units (*p* < 0.01). Pretreatment with 10 and 50 μM baicalein increased the levels of HO-1 as follows: 1.66 ± 0.04, and 1.7 ± 0.04 relative units (*p* < 0.001), respectively. Next, we determined the level of oxidative stress in Ang II-induced VSMCs. Compared with those in the Ang II-induced VSMCs, as shown in Figure 6e,f, the immunofluorescence analysis indicated that baicalein reduced the level of oxidative stress in a dose-dependent manner. The percentage of fluorescence intensity was elevated in Ang II stimulation (196.2 ± 6.59, *p* < 0.001). Pretreatment with 1, 10 and 50 μM baicalein decreased the levels of fluorescence intensity as follows: 159.7 ± 8.36; 117.3 ± 4.15; and 100.8 ± 3.57 relative units (*p* < 0.01) respectively.

## 3. Discussion

In the present study, we demonstrated that baicalein suppressed abdominal aortic aneurysm formation in Ang II-induced ApoE^−/−^ mice. Baicalein treatment significantly reduced AAA occurrence and aorta dilatation in vivo. Regarding the phenotypic characteristics of VSMCs, as per our prediction, it dose-dependently attenuated the proliferation and migration of Ang II-induced VSMCs. In particular, pretreatment with baicalein has been shown to inhibit VSMC entry into the S phase of the cell cycle via G_0_/G_1_ arrest, the stage of the cell cycle wherein DNA replication occurs. Baicalein pretreatment appears to have various modes of action in the context of Ang II-induced VSMCs. In our study, we found that baicalein pretreatment inhibits Ca^2+^ release in Ang II-induced VSMCs. These effects are due to baicalein’s ability to inhibit the expressions of PCNA, p-ERK1/2 and p-Akt in a dose-related manner. Furthermore, the phosphorylation levels of p-NF-κB and activation of the NLRP3 inflammasome were significantly increased in Ang II-induced VSMCs. Our findings show that baicalein inhibits phosphorylation of NF-κB and NLRP3 inflammasome activation. In further investigating the mechanism through which baicalein inhibits oxidative stress, we determined that it activates the expression of Nrf2/HO-1 signaling; subsequently, the production of ROS was reduced in Ang II-induced VSMCs. 

An increasing number of efforts have been made to define the protective roles of baicalein. A previous study showed that baicalein prevented AAA formation [29], while another study determined that baicalein has also been shown to significantly attenuate Ang II-induced inflammation, oxidative stress and multiple signaling pathways (Akt/mTOR, ERK1/2, NF-κB and calcineurin) in Ang II-treated mice [36]. Baicalein can promote cell survival under oxidative stress by upregulating the expression level of MARCH5 in cardiomyocytes [37]. Our experiment confirmed that Ang II increased proliferation and migration, decreased α-SMA while PCNA expression was increased in VSMCs in vitro, indicating that Ang II regulates the phenotype modulation of VSMCs; furthermore, the proliferation and migration of Ang II-induced VSMCs were inhibited by baicalein. 

Our previous findings showed that baicalein inhibited β-GP-induced VSMC calcification and apoptosis by blocking Runx-2 and BMP-2 expressions although it improves vascular contractile function by increasing the expressions of SM22α and α-SMA [38]. Resultantly, the functional role of baicalein in structural changes in the abdominal aorta still requires further investigation.

Cell cycle analysis further showed that baicalein administration increased the G_0_/G_1_ phase in VSMCs in Ang II-induced stimulation while concomitantly decreasing the population of cells in the S phase; as a result, VSMCs were unable to enter the S phase during baicalein treatment. To further examine the effects of baicalein on VSMC regulation at the nuclear level, we measured PCNA levels. PCNA is synthesized early in the G_1_ and S phases of the cell cycle and is required for cell progression from G_1_ to the S phase, so PCNA can be used as a marker of proliferating cells in normal and pathological states [39]. At this point, we demonstrated that baicalein suppressed PCNA expression and arrested cells in the G_0_/G_1_ phase, suggesting that baicalein inhibits VSMC proliferation by preventing G_1_ to S progression. These results demonstrated that baicalein inhibited PCNA levels, suggesting that the anti-proliferative effect of baicalein was a consequence of its ability to inhibit cell entry into the S phase due to interference with the early G_0_/G_1_ transition phase. 

It has been shown that extracellular-signal-regulated kinase (ERK) [40] and Akt [41] are important in the activation of inflammation in AAA formation. Our findings show that baicalein suppressed VSMC proliferation and AAA progression through inhibiting the expressions of PCNA, p-ERK and p-Akt.

Since inflammation plays an important role as a key driver of AAAs, the regulation of NLRP3 inflammasome activation could be a promising therapeutic target for AAA-related cardiovascular diseases. We therefore initially assessed the functional changes in the abdominal aorta in vivo and found that baicalein pretreatment significantly attenuated Ang II-induced vascular wall enlargement of the abdominal aorta. This indicates the attenuation effects of baicalein treatment on functional and pathological changes in the abdominal aorta in Ang II-infused mice. In addition, in the histological examination, we found that baicalein suppressed the expansion of both the luminal and outer diameters as well as medial VSMC hyperplasia and disarray of the abdominal aorta in Ang II-induced AAAs. VSMCs are major components of the vascular wall, maintaining homeostasis and structural integrity, and can transit from quiescent, differentiated cells to proliferating, migrating cells. Phenotypic changes in VSMCs in aortic aneurysms can lead to an increased expression of various genes and proteins involved in these processes, such as matrix metalloproteinases (MMPs), transforming growth factor-β (TGF-β) and cyclin D1 [42]. This can cause the loss of vascular elasticity and tension accompanied by the degeneration of elastin and collagen [43]. Regulation of phenotypic changes in VSMCs is a complex procedure which involves the integration of numerous environmental cues, that include cytokines, mechanical force, cell–cell contact, injury stimuli and, in particular, growth factor/cytokine signaling. Intracellular Ca^2+^ concentration mechanistically regulates vascular smooth muscle vessel contractility and therefore plays a critical role in the pathogenesis of cardiovascular diseases and vasoconstriction. PCNA, ERK and Akt activation has already validated as influencing VSMC differentiation. 

A growing body of evidence has supported the efficiency of phytochemicals (i.e., baicalein) in early AAA treatment [30]. Wang et al. (2016) found that baicalein prevents AAAs through the inhibition of ROS production in the aortic wall, a reduction in inflammatory cell accumulation in the aorta, downregulation of the angiotensin type 1 receptor (AT1R) and blocking mitogen-activated protein kinases (MAPKs) [29]. More recently, the involvement of baicalein in vascular protection through different mechanisms (e.g., inhibition of oxidative stress, modulation of inflammation and attenuation of vascular calcification) has been demonstrated [36]. Oxidative stress and aberrant VSMC proliferation and migration have an impact on the establishment and progression of atherosclerosis. Plaques induced by endothelial damage and inflammation are the most common pathophysiological manifestation of aneurysm development and progression (i.e., atherosclerosis). When atherosclerosis develops, pro-inflammatory stimulation and oxidative stress disrupt the normal response of VSMCs, lead to aberrant proliferation and migration while hastens the formation of aneurysms [5]. 

Vascular inflammation is a key feature in AAA formation, with the complex mechanisms of ROS production, extracellular matrix (ECM) degradation, thrombosis and hemodynamic force playing crucial roles in the pathology of AAAs [5]. To determine the mechanism through which baicalein as inhibiting AAA-induced inflammation, we treated VSMCs with and without baicalein in order to evaluate the signal transduction pathways in Ang II-induced VSMCs. The phosphorylation levels of the NF-κB and NLRP3 inflammasomes were significantly upregulated in Ang II-stimulated VSMCs, but pretreatment with baicalein effectively attenuated these effects, indicating that baicalein could suppress Ang II-induced VSMC phenotypic modulation via downregulation of the NF-κB/NLRP3 inflammasome. Based on our findings, the Ang II-infused AAA model stimulated generation of reactive oxygen species, leading to NLRP3 inflammasome aggregation and caspase-1 activation in macrophages of the vascular adventitial layer. The activated interleukin (IL)-1β is released then causing vascular inflammation and the generation of other inflammatory cytokines/chemokines, further enhancing vascular inflammation. Increased inflammation induces MMP activation and ECM degradation, leading to AAA formation [19]. In addition, this present study has shown that baicalein treatment can induce the activation of the Nrf2/HO-1 pathway in VSMCs. Nrf2 is a transcription factor that plays a key role in the regulation of antioxidant and detoxification enzymes. However, HO-1 is an enzyme that can be induced by Nrf2, and it has antioxidant and anti-inflammatory effects. Baicalein therapy can boost the expression of antioxidant and detoxifying enzymes, which can minimize the formation of ROS in VSMCs through activating the Nrf2/HO-1 pathway, while providing positive effects in protecting against oxidative stress and preventing the development of cardiovascular diseases. It has been clearly demonstrated that baicalein supresses the development of AAAs through the inhibition of VSMC proliferation via inhibiting PCNA/p-ERK/Akt pathways, increases the cell cycle arrest of VSMCs at the G_0_/G_1_ phase, supresses cellular ROS and blocks the stimulation of NF-κB/NLRP3 inflammasome pathways.

Some limitations of the current study have been identified. Initially, the study did not have a sufficiently large sample size, and the various baicalein concentrations used were not sufficent to determine its effects in preventing the development of AAAs. Furthermore, we focused on in vitro and in vivo experimental methods of treating AAAs, so these models of AAAs are unable to reflect the whole disease spectrum of human AAA pathophysiology; accordingly more clinical trials are required to rule out such limitations and provide convincing evidence that baicalein can be utilized to prevent the development of AAAs. Regardless, the current research findings provide a foundation for the discovery of new drugs to treat AAAs and also support the potential clinical application of baicalein for prevention and non-surgical treatment of AAAs in the future. 

## 4. Materials and Methods

### 4.1. Reagents

Bicalein, a 5,6,7-trihydroxy-2-phenyl-4H-1-benzopyran-4-one, was purchased from Sigma-Aldrich with the product number 465119 (Saint Louis, MO, USA). The Chemical Abstracts Service (CAS) number of baicalein was 491-67-8. The standard product is 270.24 g/mol, with a purity of no less than 98%. We dissolved the baicalein in DMSO (Sigma-Aldrich, Saint Louis, MO, USA) to make stock solutions and stored them at −20 °C. Angiotensin II was obtained from Sigma-Aldrich (product number 4474-91-3, MO, USA), having a molecular weight of 1046.18 g/mol. The trichrome stain (Masson) kit was obtained from Sigma-Aldrich (product number HT15). The elastic stain kit (Verhoeff–Van Gieson/VVG staining) was purchased from Abcam (product number ab150667, Waltham, MA, USA). Thiazolyl blue formazan (MTT) was obtained from Sigma-Aldrich (product number 88417, MO, USA). The cell proliferation kit (XTT) was purchased from Roche with the product number 11465015001 (Mannheim, Germany). Cal-520 AM was purchased from Abcam (product number ab171868, MA, USA). DCFH-DA was obtained from Invitrogen (product number D399, Waltham, MA, USA).

### 4.2. Mice

Twenty-four-week-old male apolipoprotein E-deficient mice were purchased from the Jackson Laboratory, Bar Harbor, ME, USA. Mice were apolipoprotein E knock out C57BL/6J mice. The animal study was conducted according to a previous method [44]. Prior to experiments, mice were acclimatized for one week and were maintained on a normal laboratory diet and water *ad libitum*. Cotton, shredded paper and cardboard tubes were provided as environmental enrichment for all mice, which were housed 2–3 per cage with food and tap water freely available. Mice were maintained on a 12 h light/dark cycle at a relative humidity of 55 ± 2% and a temperature of 23 ± 2 °C. The in vivo study was carried out in a pathogen-free animal facility at the Kaohsiung Medical University animal house facility, ethics approval was obtained from the Kaohsiung Medical University Animal Ethics Committee, and experiments were conducted according to the Guide for the Care and Use of Laboratory Animals guidelines published by the National Institutes of Health for scientific purposes. All methods were performed in accordance with the Animal Research: Reporting of In Vivo Experiments (ARRIVE) 2.0 guidelines [45]. 

### 4.3. Induction of AAAs

The AAA mouse model was generated according to a published method [46]. To induce AAAs, 24-week-old male apolipoprotein E-deficient mice were subcutaneously infused with 1000 ng/kg per minute for 28 days with Ang II (Sigma-Aldrich) through an osmotic pump infusion system (Alzet 2004; Durect, Cupertino, CA, USA) with the mini pump being implanted in the back of the neck of the mice. Only male mice were used owing to the possible influence of female hormonal effects on the AAA model, and the male predominance in human AAAs [47]. 

### 4.4. Control and Intervention Groups

Mice were randomly assigned to receive either control or AAA induction surgery. For sham and baicalein (BE) groups, 0.9% NaCl solution was infused for 28 days then baicalein (25 mg/kg) was intraperitoneally administered once a day starting on day 1 after pump implantation and given for 27 days. There were 4 groups: a sham group (without treatment), the BE group, the AAA group (Ang II infusion without treatment) and the AAA + BE group (Ang II infusion with BE treatment). Mice were fed a Western diet (0.15% cholesterol and 21% milk fat, 57BD; Test Diet, Richmond, IN, USA) during the study period. At study completion, mice were sacrificed by administering a phenobarbital overdose for the measurement of the suprarenal aortic diameter and collection of aortic samples on day 28 (*n* = 8 per group). 

### 4.5. Histology and Immunohistochemistry

As previously described [46,48], aortic sections were stained for collagen fibers, α-SMA and elastic degradation. Four-micrometer sections of proximal aortas were stained with Masson’s trichrome, and the collagen-positive area was expressed as the ratio of collagen fiber to muscle fiber areas. For immunohistochemistry analysis, aortic sections were used with the monoclonal anti-alpha-SMA (1:400, A2547, Sigma-Aldrich). The 4 μm paraffin sections were placed on poly-L-Lysine-coated slides, then after being deparaffinized, endogenous peroxidase was reduced via incubation with 3% hydrogen peroxidase in phosphate-buffered saline (PBS) for 5 min. Subsequently, horseradish-peroxidase-labeled goat anti-mouse IgG (1:100) was used, with 3′-Diaminobenzidine (DAB) coloration, counterstaining, dehydration, clearing and mounting being applied. To determine the stain of α-SMA, positive-stained aorta tissues were counted in 10 representative fields under an optical microscope at 40× magnification (Nikon Eclipse TE2000-S, Tokyo, Japan) and the α-SMA-positive area in each section was determined using ImageJ 1.44 software. Verhoeff–Van Gieson elastic staining was performed in the obtained samples along the entire aorta, as described previously [46,48], with each histologic stain performed in 3 inconsecutive aortic sections. For semiquantitative histology, the severity of medial elastin degradation was graded as I (mild) to IV (severe) [49].

### 4.6. Cell Culture

As previously described, VSMC cultures were obtained from the thoracic aorta of 10–12-week-old male Wistar rats [50,51]. Cells were cultured in DMEM with 10% heat-inactivated FBS, 100 U/mL penicillin and 100 g/mL streptomycin and were maintained at 37 °C in a humidified 5% CO_2_ incubator. To confirm the purity of VSMC cultures, immunocytochemical localization of smooth muscle actin was conducted, and when the cultures reached confluence, cells were subcultured using 0.5% (*w*/*v*) trypsin. Substitutions of culture media were made every 3 days, and passage numbers from 3 to 6 were used for the experiment.

### 4.7. Cell Number

Cell count experiments were performed as described previously [50]. A suspension of VSMCs (5 × 10^4^ cells/well) was pretreated with or without baicalein (1, 10, 50 and 100 μM) for 1 h and then incubated with or without Ang II (0.1 μM) for 24 h. Cells were counted in a hemocytometer under a light microscope.

### 4.8. Cell Viability and Cell Proliferation Assay

The MTT and XTT assays were generated as described in our previous studies [50,51]. Following a simultaneous process of 24 h serum deprivation, baicalein was used to incubate VSMCs at different concentration levels for 24 h; subsequently, MTT solution (0.5 mg/mL) was added and incubated for 4 h at 37 °C. After the MTT solution was removed, isopropanol was added, and the cells were shaken for 10 min, then MTT formazan crystals were quantified by determining the absorbance at 540 nm and 630 nm respectively, using an enzyme-linked immunosorbent assay (ELISA) reader (DYNEX Technologies, Denkendorf, Germany). The indirect marker of cell proliferation was performed by measuring metabolic activity through the XTT test as per the manufacturer’s instructions (Roche Molecular Biochemicals, Mannheim, Germany).

### 4.9. Wound Healing Assay

Cells were plated in 12-well cell culture plates with cell growth media containing FBS. When the cells reached semi-confluence, FBS was removed and replaced with serum-free media. To produce a clean wound area, a straight line was made by scratching the center of the plate with a plastic pipette tip 24 h after serum depletion.

Numerous images of the wound were taken 24 h post-wounding in serum-free medium (control) or in the presence of Ang II (0.1 μM) with the effect of baicalein being evaluated by adding it to the culture medium for 1 h before adding Ang II. For comparison, the migration distance between the edge of the wound and the leading edge of the migrating cells was measured, with all images being captured using a Nikon TE2000-S microscope (Tokyo, Japan).

### 4.10. Boyden Chamber Assay

As previously explained, VSMC migration was assessed on Transwell polyethylene terephthalate cell culture inserts with 8 μm pores [50,51]. Ang II (0.1 μM) was dissolved in DMEM and placed in the lower compartment either in the presence or absence of baicalein (1–100 μM), then VSMCs (2 × 10^4^ cells) were loaded into the upper compartment and incubated for 24 h at 37 °C and 5% CO_2_/95% air. The cells on the upper membrane surface were removed, and those on the lower surface were fixed in methanol and stained with Giemsa. Next, the numbers of cells per six high-power fields (200× HPF) were counted with the mean number of cells was used to express migration activity; finally, captured the images were captured with a Nikon TE2000-S microscope (Japan).

### 4.11. Flow Cytometry Assay of Cell Cycle

Via serum depletion for 48 h, VSMCs were synchronized to the G0 phase. The cells were pre-incubated with different concentrations of baicalein (1–100 μM) after being replenished with new DMEM, and Ang II (0.1 μM) was added to promote cell cycle advancement. Cells were trypsinized after 48 h and then centrifuged at 1250× *g* for 5 min, washed twice with cold PBS and then treated with RNase A (10 μg/mL). Propidium iodide (50 g/mL) was used to stain the DNA for 30 min at 37 °C, and 5 × 10^4^ cells were then examined via flow cytometry (Becton, Dickinson and Company, Franklin Lakes, FL, USA). We captured the cells using the Modifit LT 4.1 software and then calculated the percentage of cells in the G0/G1, S and G2/M phases of the cell cycle respectively [50,51].

### 4.12. Measurement of Calcium Concentration in VSMCs 

VSMCs were cultured in a confocal dish at a density of 5 × 10^4^ cells/well and were treated with or without baicalein (1, 10, 50 and 100 μM) for 1 h. At the end of treatment, Cal-520 AM (5 mM, ab171868) was diluted with Ca^2+^-free Hanks’ balanced salt solution (HBSS) in the dish. After incubation at 37 °C in the dark for 1 h, cells were incubated in Ca^2+^-free HBSS and then Ang II (0.1 μM) was added for 6 h. Fluorescence images were taken with a fluorescence microscope (Nikon, TE2000-S, Japan) at 200× magnification in real time. The fluorescence intensity of intracellular calcium was measured using Carl Zeiss AxioVision 4.8 software.

### 4.13. Western Blot Analysis

After being put into a state of dormancy for 48 h, VSMCs were stimulated with Ang II (0.1 μM) before being incubated for 1 h in either the absence or presence of baicalein. Cells were collected after the reactions were stopped via washing them twice with cold PBS. Briefly, total cell extracts were prepared using a protein lysis buffer, and then the experiment was placed on 10% SDS-polyacrylamide electrophoresis gels, which were then transferred to PVDF membranes, blocked and then incubated them with primary antibodies: a 1:1000 dilution of anti-PCNA (#2586, Cell signalling, Danvers, MA, USA), anti-phospho-ERK 1/2 (#9101, Cell signalling), anti-ERK 1/2 (#9102, Cell signalling), anti-phospho-Akt (GTX128414, GeneTex, Irvine, CA, USA), anti-Akt (GTX121937, GeneTex, CA, USA), anti-NLRP3 (tcba2712, Taiclone, Taipei, Taiwan), anti-PYCARD (ASC; tcba1690, Taiclone), anti-caspase-1 (22915-1-AP, proteintech, Rosemont, IL, USA), anti-IL-1β (sc-7884, Santa Cruz, Dallas, TX, USA), anti-Keap1 (10503-2-AP, proteintech), anti-Nrf2 (GTX103322, GeneTex), anti-HO-1 (GTX101147, GeneTex) and anti-β-actin (GTX629630, GeneTex) antibodies. The membranes were treated with the appropriate secondary antibodies (1:1000; Merck Millipore, Burlington, MA, USA) for 1 h and then exposed to photographic film after being incubated with the enhanced chemiluminescence (ECL) reagents (Merck Millipore, Darmstadt, Germany) to determine protein expressions.

### 4.14. Immunofluorescence Staining of NF-κB Nuclear Translocation 

VSMCs were cultured in a confocal dish at a density of 5 × 10^4^ cells/well and incubated with Ang II (0.1 μM) in the presence or absence of baicalein (50 μM) for 6 h. At the end of treatment, fixation in 4% paraformaldehyde at room temperature for 30 min was carried out, followed by permeabilization with 0.1% Triton X-100 for 5 min at 4 °C. Cells were incubated with anti-phospho-NF-κB p65 (#3033, Cell signalling; 1:100) antibody overnight. Subsequently, cells were incubated with the anti-rabbit IgG tagged with AlexaFluor 488 for 1 h and stained them with DAPI for 10 min. Finally, the images of NF-κB nuclear translocation were captured with a confocal microscope coupled with an image analysis system (Olympus Fluoview FV1000) at 600× magnification.

### 4.15. Determination of Intracellular Reactive Oxygen Species

The determination of ROS was accomplished with fluorescent stain, DCFH-DA. Briefly, VSMCs (5 × 10^4^ cells) were incubated with Ang II (0.1 μM) in the presence or absence of baicalein (1–50 μM) for 6 h. In order to measure the ROS production induced by Ang II, cells were stained with 10 μM DCFH-DA for 30 min at 37 °C and then observed under a fluorescence microscope (Nikon, TE2000-S, Japan) with fluorescence intensity of intracellular ROS production measured using the Carl Zeiss AxioVision 4.8 software.

### 4.16. Statistical Analyses

Data were presented as mean ± S.E.M from at least five independent experiments for each group. The analysis of variance (ANOVA) technique, followed by the Tukey post hoc test, was used to establish the statistical significance of the differences; a value of *p* < 0.05 was regarded as significant.

## 5. Conclusions

Based on the present study, we confirmed that baicalein pretreatment suppresses Ang II-induced AAA formation in mice, particularly via VSMC phenotypic alleviation of oxidative stress by activating the NF-κB/NLRP3 inflammasome signaling pathways (Figure 7). Our findings provide further evidence for a preventive role of baicalein in AAAs although more studies are required for complete elucidation, in particularly revealing the interplay between environmental stimuli and signaling pathways within VSMCs.

## Figures and Tables

**Figure 1 ijms-24-16004-f001:**
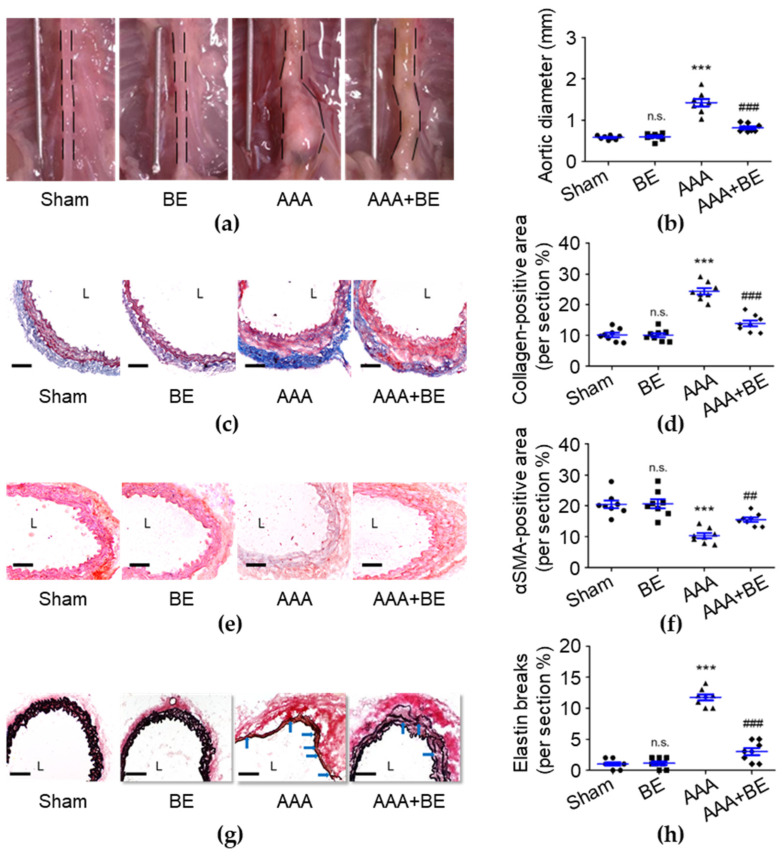
Baicalein treatment alleviated Ang II-infused abdominal aortic aneurysm formation. (**a**) Representative photos of the aortas. (**b**) Maximal aortic diameter measurement (*n* = 8 per group). (**c**) Representative microscopic images of Masson’s trichrome staining. (**d**) Collagen-positive area (*n* = 8). (**e**) Representative microscopic images of α-SMA staining. (**f**) α-SMA-positive area (*n* = 8). (**g**) Representative microscopic images of Verhoeff–Van Gieson’s staining. (**h**) Elastin break number (*n* = 8). BE indicates baicalein; AAA indicates abdominal aortic aneurysm; L indicates lumen; blue arrows indicate disrupted elastic lamellae. All scale bars represent 50 μm. Data are expressed as mean ± standard error of mean (SEM). Non-significant (n.s.) *p* > 0.05 and *** *p* < 0.001 compared with the sham group. ^##^
*p* < 0.01 and ^###^
*p* < 0.001 compared with the AAA group.

**Figure 2 ijms-24-16004-f002:**
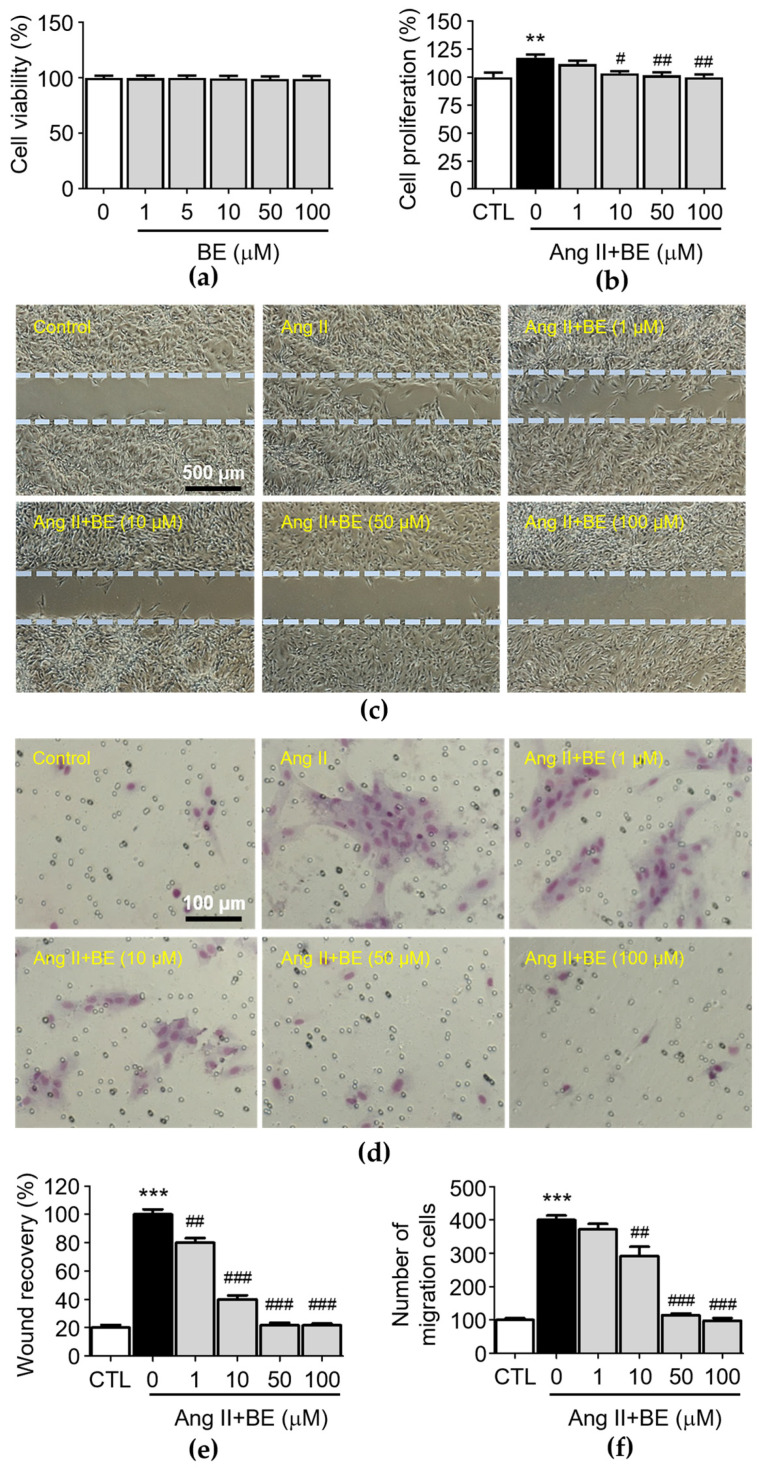
Baicalein treatment inhibited Ang II-induced VSMC proliferation, motility and migration. The cells were cultured with the indicated concentrations of baicalein with or without Ang II for 24 h. (**a**) The cell viability was determined using an MTT assay (*n* = 6). (**b**) The cell proliferation was determined using an XTT assay (*n* = 6). (**c**) Representative microscopic images of wound healing assay. (**d**) Representative microscopic images of Giemsa staining (purple). (**e**) The wound area of VSMCs was quantified (*n* = 6). (**f**) The cell migration of VSMCs per field was measured (*n* = 6). Data are expressed as mean ± SEM. ** *p* < 0.01 and *** *p* < 0.001 compared with the control group. ^#^
*p* < 0.05, ^##^
*p* < 0.01 and ^###^
*p* < 0.001 compared with the Ang II-only group.

**Figure 3 ijms-24-16004-f003:**
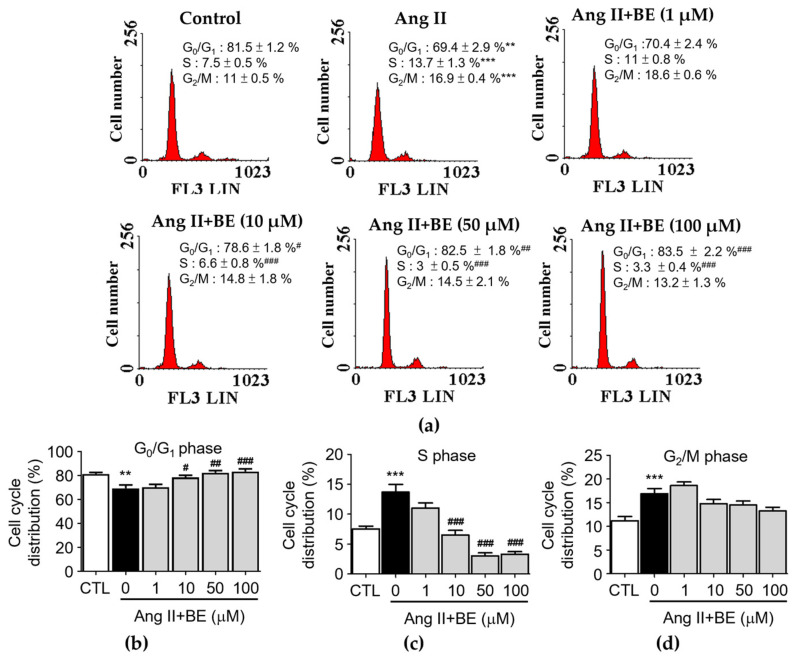
Baicalein treatment inhibited entering the S phase of the cell cycle in Ang II-induced VSMCs. The cells were cultured with the indicated concentrations of baicalein with or without Ang II for 24 h. (**a**) PI staining assay was performed on the cell cycle distribution via flow cytometry (red). (**b**–**d**) Statistical analysis of cell cycle phase distribution. G_0_/G_1_ phase (**b**), S phase (**c**) and G_2_/M phase (**d**) are shown (*n* = 6). Data are expressed as mean ± SEM. ** *p* < 0.01 and *** *p* < 0.001 compared with the control group. ^#^
*p* < 0.05, ^##^
*p* < 0.01 and ^###^
*p* < 0.001 compared with the Ang II-only group.

**Figure 4 ijms-24-16004-f004:**
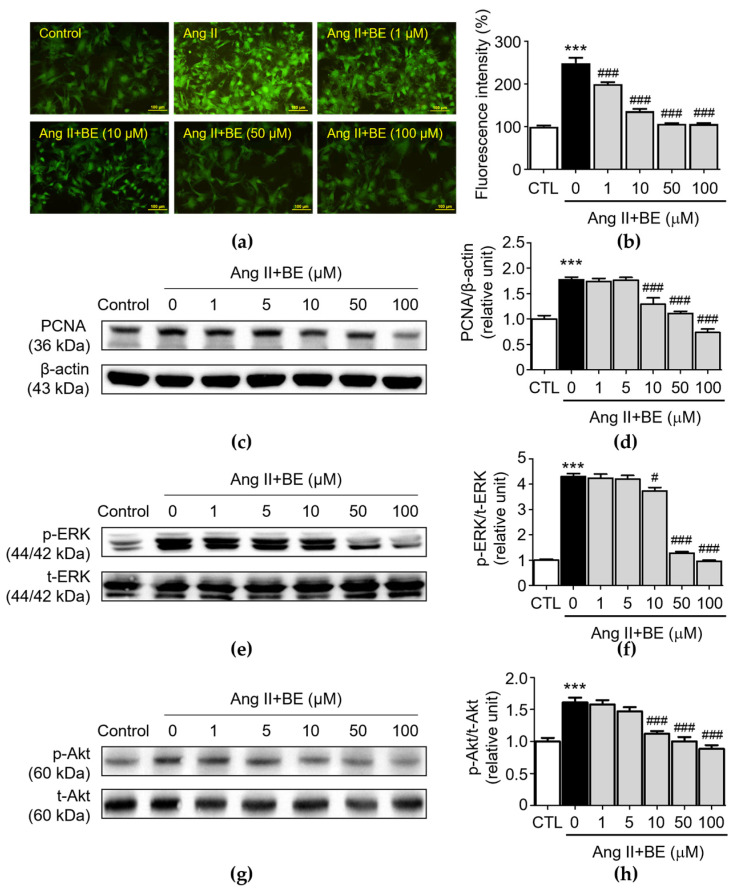
Baicalein treatment reduced Ca^2+^ release, PCNA, ERK and Akt activation in Ang II-induced VSMCs. The cells were cultured with various concentrations of baicalein with or without Ang II for the indicated times. (**a**) Representative microscopic images of Cal-520 AM staining (green). (**b**) Quantitative analysis of the relative fluorescence intensity at each field (*n* = 6). (**c**–**h**) The protein expressions of PCNA (**c**,**d**), ERK (**e**,**f**) and Akt (**g**,**h**) were analyzed via Western blotting and then quantitatively measured (*n* = 5). All scale bars represent 100 μm. Data are expressed as mean ± SEM. *** *p* < 0.001 compared with the control group. ^#^
*p* < 0.05 and ^###^
*p* < 0.001 compared with the Ang II-only group.

**Figure 5 ijms-24-16004-f005:**
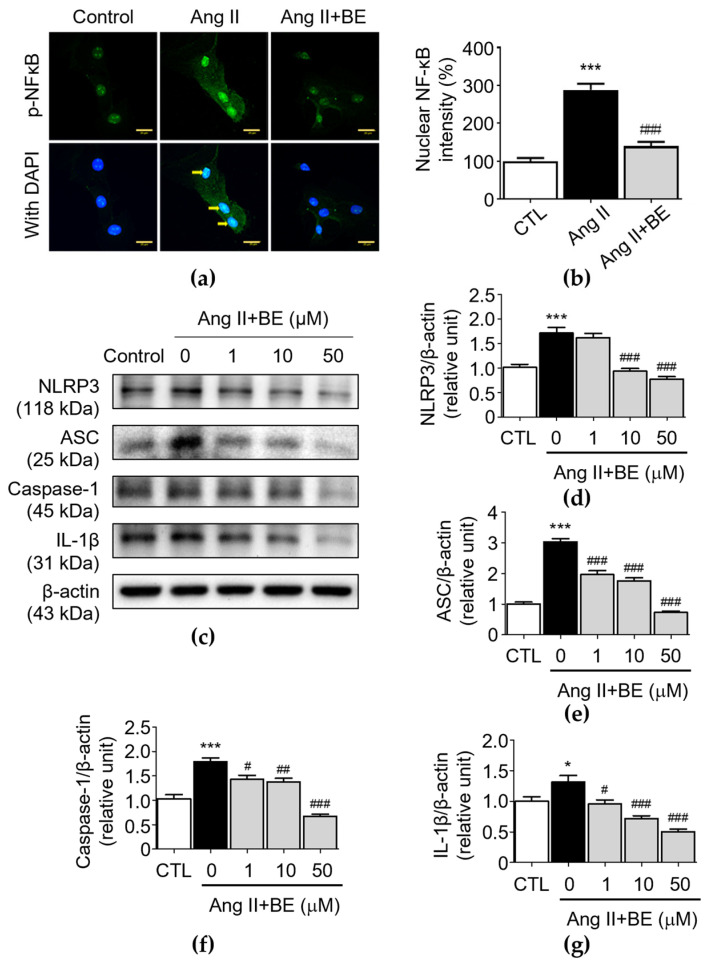
Baicalein treatment inhibits NF-κB and NLRP3 inflammasome pathways in Ang II-induced VSMCs. The cells were cultured with various concentrations of baicalein with or without Ang II for 24 h. (**a**) Representative microscopic images of NF-κB immunofluorescence staining. Yellow arrows show NF-κB nuclear translocation (**b**) Quantitative analysis of the relative nuclear NF-κB fluorescence intensity at each field (green) (*n* = 6). (**c**–**g**) The protein expressions of NLRP3 (**c**,**d**), ASC (**c**,**e**), caspase-1 (**c**,**f**) and IL-1β (**c**,**g**) were analyzed by Western blotting and then quantitatively measured (*n* = 5). All scale bars represent 20 μm. Data are expressed as mean ± SEM. ** p* < 0.05 and **** p* < 0.001 compared with the control group. ^#^
*p* < 0.05, ^##^
*p* < 0.01 and ^###^
*p* < 0.001 compared with the Ang II-only group.

**Figure 6 ijms-24-16004-f006:**
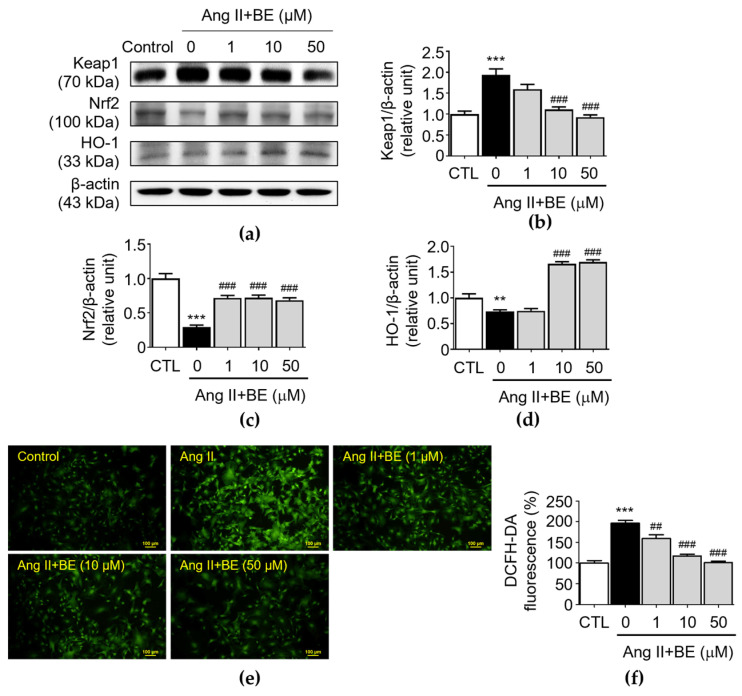
Cells were cultured with various concentrations of baicalein with or without Ang II for 24 h. (**a**–**d**) The protein expressions of Keap1 (**a**,**b**), Nrf2 (**a**,**c**) and HO-1 (**a**,**d**) were analyzed via Western blotting and then quantitatively measured (*n* = 5). (**e**) Representative microscopic images of DCFH-DA staining (green). (**f**) Quantitative analysis of the relative fluorescence intensity at each field (*n* = 6). All scale bars represent 100 μm. Data are expressed as mean ± SEM. *** p* < 0.01 and **** p* < 0.001 compared with the control group. ^##^
*p* < 0.01 and ^###^
*p* < 0.001 compared the Ang II-only group.

**Figure 7 ijms-24-16004-f007:**
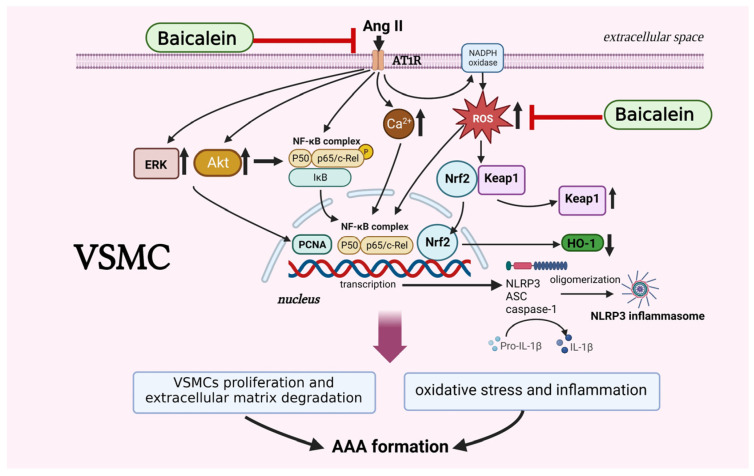
The proposed mechanisms of baicalein treatment suppress proliferation, motility, migration and Ca^2+^ release in Ang II-induced VSMCs. Baicalein treatment ameliorates Ang II-induced inflammation and ROS production, leading to the downregulation of NLRP3 inflammasome pathway. Treatment with baicalein, in line with results, leads to inhibition of Ang II-infused AAA formation in mice. The symbol “

” means inhibition, symbol “↑” means increase, and symbol “↓” means decrease. The picture was created with BioRender.com, access date (6 May 2023).

## Data Availability

Not applicable.

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
