# Peer review of "Vasculoprotective Potential of Baicalein in Angiotensin II-Infused Abdominal Aortic Aneurysms through Inhibiting Inflammation and Oxidative Stress"

_ijms, 2023, doi:10.3390/ijms242116004_

Round 1
Reviewer 1 Report
Comments and Suggestions for Authors
Abstract:
- The abstract provides a concise overview of the study. However, it lacks specific information about the methodology employed, such as the animal model used and the duration of the study.
Introduction: 2. The introduction is well-written and provides a good background on abdominal aortic aneurysm (AAA). However, it could benefit from a more precise statement of the research hypothesis or objectives.
-
The introduction mentions common risk factors for AAA but does not specify the source of this information (references) or provide specific statistics. It would be helpful to include supporting references for these statements.
-
The introduction discusses the role of inflammation in AAA but does not reference specific studies or evidence supporting this claim. It would enhance the scientific credibility of the introduction to include relevant citations.
-
There is an inconsistency in the use of terminology. In some places, "baicalein" is mentioned before its full name, while in others, it is introduced with its full chemical name. It would be more consistent to refer to it consistently throughout the introduction.
6. The methods section should include details about the animal model used, including species, strain, and age. Information about the administration of Angiotensin II and baicalein, dosages, and the duration of treatment should be included.
Results: 7. The results section provides a summary of the key findings. However, it lacks specific data and statistical analysis, which are essential for scientific reporting. Including actual numerical results and statistical significance would strengthen this section.
-
Figures 1 and 2 are mentioned in the text, but the text does not provide a description of what each figure represents. Each figure should be described and discussed in detail in the results section.
-
Figure 3, Figure 4, and Figure 5 are referenced in the text but are not included in this document. It is crucial to include these figures for a complete understanding of the results.
Discussion: 10. The discussion section could benefit from a more in-depth analysis and interpretation of the findings. For example, why is G0/G1 cell cycle arrest significant, and how does it relate to the pathogenesis of AAA?
-
The discussion mentions the inhibition of NFκB and NLRP3 inflammasome activation but does not explore the potential mechanisms through which baicalein exerts these effects. Providing insights into the underlying molecular mechanisms would enhance the discussion.
-
The discussion should also address the limitations of the study, potential sources of bias or error, and avenues for future research in this area.
6: -
Materials and Methods:
-
Reagents:
- Mention the CAS (Chemical Abstracts Service) number for baicalein.
- Clarify the source or method used to determine the purity of baicalein (stated as "no less than 98%").
-
Ang II-infused AAA model and baicalein treatment:
- Provide more details on the source and handling of apolipoprotein E-deficient mice.
- Specify the composition of the Western diet.
- Describe the specific guidelines and regulations followed for animal experiments.
-
Histology and Immunohistochemistry:
- Explain the grading criteria for medial elastin degradation (graded as I to IV).
-
Cell culture:
- Detail the cell culture conditions, including DMEM type, FBS source, and incubation conditions.
- Clarify the purpose and methodology of immunocytochemical localization of smooth muscle actin.
-
Cell number:
- Specify the units for cell counts (e.g., cells per mL or cells per well).
-
Cell viability and cell proliferation assay:
- Explain the significance of measuring absorbance at both 540 nm and 630 nm for MTT formazan crystals.
- Provide specifics on following the manufacturer's instructions for the XTT test.
-
Wound healing assay:
- Describe the method used to measure migration distance and potential sources of error.
-
Boyden chamber assay:
- Detail the composition of DMEM in the lower compartment of transwell inserts.
-
Flow cytometry assay of cell cycle:
- Explain the method used to determine the percentage of cells in each phase of the cell cycle and the software for analysis.
-
Measurement of calcium concentration in VSMCs:
- Provide details on the method for measuring calcium concentration and calibration procedures.
- Clarify whether fluorescence intensity values were normalized or compared to controls.
-
Western blot analysis:
- Specify conditions for protein lysis and lysis buffer composition.
- Describe the method used for quantifying protein expression and how band intensity was measured.
-
Immunofluorescence staining of NFκB nuclear translocation:
- Explain the rationale behind selecting the specific concentration of baicalein (50 μM).
-
Determination of intracellular reactive oxygen species:
- Clarify if DCFH-DA staining assessed the effect of baicalein on ROS induced by Ang II or ROS levels in general.
- Provide more details on how fluorescence intensity was quantified.
-
Statistical analyses:
- Specify the number of replicates or samples used for each experiment.
English needs to be improved
Author Response
We deeply appreciate the expertise comments to improve this manuscript from the reviewers. Therefore, we have revised sentences as their suggestions. Furthermore, we provided a point-by-point responses to the comments of reviewers.
Please see the attachment.

Reviewer 2 Report
Comments and Suggestions for Authors
The authors further investigate the protective effects of baicalein reported in literature on aortic aneurism with in vivo and in vitro models. The paper is well written (apart some strange sentences, see below) and data support the conclusion drawn apart for some specific points that need to be addressed before publication.
Major points:
The angiotensin II based AAA model used is correctly described in material and methods section but no reference are provided to evaluate whether is a relevant model for human AAA and this is not discussed. Moreover, it features reduced alphaSMA upon angII induction (that is a marker VSMC) which is reduced by baicalein treatment. Thus, is very unclear its connection with in vitro Angiotensin II induced proliferation of VSMC and baicalein induced decrease in it. Moreover, decreasing VSMCs calcium mediated contraction and proliferation would not weaken arterial wall? All of this need to be better discussed.
I get really surprised by Fig 2a and 2b reporting MTT and XTT giving different results as (despite sold as viability and proliferation kits respectively) are very similar molecules both measuring mitochondrial respiratory potential. The main difference between XTT assay and MTT assay is the solubilization step (unlike MTT, XTT is reduced to a highly water-soluble orange-colored product after reduction by mitochondrial enzymes that are only present in metabolically active live cells instead of the insoluble formazan product formed from MTT). Please explain this discrepancy. Moreover, where cells starved before proliferation experiments or this is proliferation in addition to standard growth media? In Fig, 3 a starvation is reported as required before Ang II treatment.
“In contrast, baicalein treatment had non-significant influence in G2/M phase (Figure 3d).” This would be difficult to explain (a decrease in S phase with no difference in G2/M). Have you tried longer incubation time to allow cells in S phase to enter G2/M?
Is the reduction of oxidative stress the mechanism of baicalein action? A reducing agent in the medium would equally perform? Is AngII receptor downregulated by baicalein? Measure it! If this is the case, as suggested by the discussion, it offers an easy mechanism for the observations reported as all is based on AngII, also the in vivo model.
Are the baicalein concentration used achievable in vivo? This should be discussed in view of practical applications.
Minor points
Fig. 5a should be equalized for staining intensity before quantification.
262 “To clarify the connection between Nrf2 and Keap-1, cell lysates were immunoblotted with their antibodies, respectively.” I do think that this experiment measures the effect of Ang II and BE on Nrf2 and Keap-1 not their reciprocal connection.
113- 114 “Experimental in vivo and in vitro studies suggest the potential benefits of baicalein in preventing or reducing the progression of AAA.” STRONGLY NEED REFERENCE
115-117 “Moreover, this extensively utilized natural product has been demonstrated to work via a variety of processes, including gene expression control, metabolism and cell signaling pathways” STRONGLY NEED REFERENCE
233 “To demonstrate the effectiveness of baicalein at inhibiting the inflammation, VSMCs were exposed to Ang II, a well-characterized activator of NFκB.” STRONGLY NEED REFERENCE
208 “To further validate the regulatory effects of baicalein treatment on both calcium deposition and biomarkers of VSMCs proliferation, our current study isolated primary VSMCs, which was confirmed by fluorescence staining for intracellular calcium assay and western blotting for protein expression analysis.” So the previous experiments were done on cell lines? What was confirmed by calcium assay?
144 (3 ± 0.57%) here and in all manuscript carefully check the number of significative digits
PLEASE carefully revise unclear sentences:
87-89 “Caspase-8 and FAS-mediated death domain protein (FADD), and NOD1/2 are associated during
the priming phase by regulating the NFκB pathway.”
90-91 “The activation signal, or is named with signal 2”
190 “With the percentage of Ang II-induced cells in the G0/G1 phase reduced;”
Comments on the Quality of English LanguageSome sentences need to be revised
Author Response

(The authors gave the same response as above.)

Round 2
Reviewer 2 Report
Comments and Suggestions for Authors
Good revision although some minor issues about discussion persist. Pay attention to a comma in row 183
Comments on the Quality of English LanguageSome issues about language persist.